# Microalgae Encapsulation Systems for Food, Pharmaceutical and Cosmetics Applications

**DOI:** 10.3390/md18120644

**Published:** 2020-12-15

**Authors:** Marta V. Vieira, Lorenzo M. Pastrana, Pablo Fuciños

**Affiliations:** Food Processing and Nutrition Group, International Iberian Nanotechnology Laboratory, Av. Mestre José Veiga s/n, 4715-330 Braga, Portugal; marta.vieira@inl.int (M.V.V.); lorenzo.pastrana@inl.int (L.M.P.)

**Keywords:** bioactive, drug delivery systems, functional food, cosmeceuticals

## Abstract

Microalgae are microorganisms with a singular biochemical composition, including several biologically active compounds with proven pharmacological activities, such as anticancer, antioxidant and anti-inflammatory activities, among others. These properties make microalgae an interesting natural resource to be used as a functional ingredient, as well as in the prevention and treatment of diseases, or cosmetic formulations. Nevertheless, natural bioactives often possess inherent chemical instability and/or poor solubility, which are usually associated with low bioavailability. As such, their industrial potential as a health-promoting substance might be severely compromised. In this context, encapsulation systems are considered as a promising and emerging strategy to overcome these shortcomings due to the presence of a surrounding protective layer. Diverse systems have already been reported in the literature for natural bioactives, where some of them have been successfully applied to microalgae compounds. Therefore, this review focuses on exploring encapsulation systems for microalgae biomass, their extracts, or purified bioactives for food, pharmaceutical, and cosmetic purposes. Moreover, this work also covers the most common encapsulation techniques and types of coating materials used, along with the main findings regarding the beneficial effects of these systems.

## 1. Introduction

Microalgae are a heterogeneous group of photosynthetic microorganisms, whose evolutionary and phylogenetic diversity has provided a vast assortment of biochemical compositions [1]. These microorganisms are able to biosynthesize, accumulate and secrete a great range of primary and secondary metabolites as a response to changes in the external environment, many of which are highly valuable substances with industrial applications and health benefits [2].

The use of microalgae by humans dates back thousands of years, where it was used as a food source by different populations; nevertheless, the commercial exploitation of this resource is only a few decades old, when there was an apprehension regarding a possible insufficient protein supply due to the rapid increase in the world population [3,4]. Microalgae are well-known for their high protein and nutritional content, but more recently, studies have been focused on the unique biologically active compounds produced by their species, such as polyunsaturated fatty acids, pigments, antioxidants, polyphenols, polysaccharides, and other equally important substances [3,4].

Lately, there has been a growing trend towards using natural ingredients in food, pharmaceutical, and cosmetic industries due to the increasing concerns regarding consumer safety, environmental sustainability, and regulatory issues over the introduction of synthetic chemicals in human nutrition, healthcare, and beauty products [5,6,7]. Several microalgae bioactives possess significant biological activities, including anticancer, antioxidant, anti-inflammatory, antimicrobial, and immunomodulatory activities, among others [8,9,10]. Therefore, the use of such compounds seems to be a promising and innovative approach to the development of healthier, functional, and sustainable products.

Overall, microalgae may be proposed to obtain commodities with existing market value, refined bioactives, or the whole cell could even be the target product [11]. Yet, purified compounds or bioactive extracts are usually chemically unstable and strongly susceptible to oxidative degradation, particularly when exposed to oxygen, light, moisture, extreme pH, and high temperatures. The oxidative degradation may also deteriorate the compounds, leading to the development of unpleasant tastes and off-odours in the fortified product and, subsequently, may result in a negative effect on shelf stability, sensory characteristics, and consumer acceptability of the product [12]. Moreover, the low bioavailability and poor water solubility are usually recurrent issues related to the application of microalgae bioactives; in pharmaceutical and functional food products, for instance, the absorption of these compounds may be hindered due to gastrointestinal tract conditions, as well as due to their physicochemical properties [13,14].

These developmental and technological issues address the importance of researching strategies to preserve the functionality of microalgae bioactives from processing until they reach their target site. In this context, encapsulation systems are considered a promising approach, which have been applied successfully in diverse fields. The process of encapsulating a bioactive consists of its entrapment within one or more coating materials through different techniques, resulting in nano- or microparticles [15]. This strategy is associated with several advantages, including protecting the bioactive compound during processing, storage, and distribution; promoting release control; masking off-flavours; improving solubility and bioavailability, among others [16,17].

Considering the above mentioned, the present review aimed at describing the encapsulation systems reported in the literature of different microalgae biomass, its extracts, or purified compounds, focusing on food, pharmaceutical, and cosmetic applications. The described techniques of encapsulation, types of coating material, and the main findings regarding the beneficial effects of these systems were also considered.

## 2. Microalgae

Microalgae are single-celled, ubiquitous, prokaryotic, and eukaryotic primary photosynthetic microorganisms, which are taxonomically and phylogenetically diverse [9,18]. They are ancestral living organisms that have adapted uniquely to extreme habitats over billions of years of evolution and can be found almost anywhere on Earth; in freshwater, seawater, and hypersaline environments, but also in moist soils and rocks [19]. Their classification is based on various properties, such as pigmentation, the chemical nature of photosynthetic storage products, the organization of photosynthetic membranes, and other morphological features. The most abundant microalgal classes are Cyanophyceae (blue-green algae), Chlorophyceae (green algae), Bacillariophyceae (including the diatoms), and Chrysophyceae (including golden algae) [3,20]. A resume of the main microalgae classes, their most studied species, and associated biological activities are described in Figure 1.

Interest in microalgae cultivation has been prospering globally in recent decades for diverse reasons. There are several industrial and commercial applications associated with these microorganisms and examples of success include formulations in different sectors, such as functional foods, feed, cosmetics, pharmaceuticals, and fertilizers; as well as tools for wastewater treatment and biofuel production [21,22]. Moreover, many advantages have already been reported involving their cultivation process in comparison with other feedstocks. Firstly, microalgae reproduce themselves using photosynthesis to convert sun energy into chemical energy, completing an entire growth cycle every few days. Secondly, they can grow almost anywhere, requiring mostly sunlight and some simple nutrients; although the process can be accelerated heterotrophically by the addition of specific nutrients and changes in cultivation parameters. Accordingly, microalgae have much higher growth rates and productivity when compared to conventional forestry, crops, and other aquatic plants, demanding much less land area [23,24].

Through adaptive evolution and metabolic diversity, microalgae have developed a wide range of high value biologically active compounds, comprising pigments, antioxidants, polysaccharides, triglycerides, fatty acids, and vitamins [25]. It is estimated there are 70,000 to one million microalgae species; however, only about 44,000 have already been described. Furthermore, from those, only a limited number have been studied for commercial purposes [26]. Some of the most biotechnologically relevant microalgae are the green algae (Chlorophyceae) *Chlorella vulgaris*, *Haematococcus pluvialis*, *Dunaliella salina,* and the Cyanobacteria *Arthrospira platensis*, which are broadly commercialized, mainly as nutritional supplements for humans and as animal feed additives [27].

The multicellular filamentous Cyanobacteria from the genus *Arthrospira* (formerly known as “Spirulina”) occur naturally in alkaline lakes and ponds, being widely cultured around the world. The two most important species of *Arthrospira*, *A. maxima,* and *A. platensis*, are commonly applied both as a functional ingredient in food preparations and as a source of the blue photosynthetic pigment C-phycocyanin, which is used in cosmetics and the food industry [28]. This species has been used as a nutrient-rich (especially vitamin B_12_ and proteins) food source with the oldest records indicating use by the Aztecs, who harvested this microalga from Lake Texcoco in Mexico; and by the local people in Lake Chad, who used *A. platensis* as a nutritional supplement known as “dihe” [6,29]. Apart from its significance as a food additive, *A. platensis* is also recognized by the broad range of potential medical and pharmaceutical applications attributed to its metabolites. Studies have evidenced several biological activities, such as antitumor, antibacterial, anti-inflammatory and hepatoprotective activities, among others, directly related to the antioxidant capacity ascribed for C-phycocyanin and other compounds [30,31]. Similarly, the freshwater unicellular blue-green microalga *Aphanizomenon flos-aquae*, which grows spontaneously in Upper Klamath Lake in Oregon, USA, is also consumed as a nutrient-rich food source and for its health properties. Similar to the *Arthrospira* species, *A. flos-aquae* is an important source of the pigment C-phycocyanin; hence, demonstrating a strong antioxidant potential [32,33].

The unicellular green alga *Chlorella* is one of the largely studied microalgae due to their biotechnological importance as a valuable source of nutrients. Species from this genus were one of the first microalgae considered for mass cultivation and the first microalga produced commercially. *Chlorella* cells actively growing under normal conditions are typically rich in protein (40–60%) and are largely made up of essential amino acids, with a profile that suits human nutrition. In this context, *Chlorella* biomass may be considered as a desirable candidate for protein supplements or single-cell protein [34,35]. Furthermore, these species are also rich in carotenoids, vitamins and other bioactives, demonstrating potential health benefits, such as efficacy on gastric ulcers, wounds, and constipation; preventive action against both atherosclerosis and hyper-cholesterol; and antitumor activity. The suggested most important active compound is β-1,3-glucan, which is believed to be an active immune-stimulator, free radical scavenger, and a reducer of blood lipids [4,36,37].

Another important member of the green algae class is the freshwater unicellular microalga *Haematococcus pluvialis*. Under extreme environmental conditions, such as high-intensity light or oligotrophic circumstances, this species undergoes several morphological and biochemical modifications, including an intense biosynthesis of the carotenoid astaxanthin [38]. In the last few decades, *H. pluvialis* has received significant attention from the scientific and biotechnological communities for being considered as the most significant biological source of that carotenoid in nature [39,40]. Astaxanthin is a natural pigment with several applications in the nutraceutical, cosmetic, food, and feed industries [41]. Moreover, it also possesses a powerful antioxidant potential due to its unique chemical configuration, which is associated with assorted biological activities demonstrated in both animal and clinical studies [42]. Many authors have already described astaxanthin’s valuable effects in inflammatory responses and the immune system, in hypertension, cancer, ocular and cardiovascular diseases, as well as in skin ageing defence [43,44,45].

Regarding carotenoid production, the unicellular green microalga *Dunaliella salina* is equally important for its recognition as the richest source of natural β-carotene [46]. When exposed to specific extreme environmental conditions, such as high-intensity light, high salinity, extreme temperatures, and/or nutrient deprivation, *D. salina* can accumulate an exceptionally large amount of β-carotene (up to 14% of the dry algal biomass), resulting in orange-coloured cells. This great carotene productivity has led to the large-scale application of *D. salina* for commercial production of natural β-carotene, widely used as an antioxidant and colourant in the food, feed, cosmetics, and pharmaceutical industries [47,48]. Additionally, this species also contains other important lipid components, glycerol, proteins, and carbohydrates [49].

In the context of microalgae importance in different fields, some species have not been fully explored, but have been demonstrated to be a promising source of bioactives according to published studies. The microalga *Phaeodactylum tricornutum* is a marine diatom, which accumulates eicosapentaenoic acid (EPA, 20:5n-3) as a major component of its fatty acid content [50]. This species is also a rich source of the carotenoid fucoxanthin, whose intake has been suggested to improve insulin resistance and to decrease the blood glucose level, along with anticancer and anti-inflammatory effects [51,52]. Furthermore, some other microalgae genera, such as *Nannochloropsis*, *Tetraselmis, Scenedesmus,* and *Isochrysis*, have revealed their importance due to the production of long-chain fatty acids, i.e., docosahexaenoic acid (DHA) and EPA, representing also a source of antioxidant compounds [53,54].

Although there are a very large number of red algae (Rhodophyta) in nature, only a few species represent the microalgae group. The genus *Porphyridium* is the most studied one due to the particular interest in its species as a source of sulphated polysaccharides, proteins, the polyunsaturated fatty acids (PUFAs) arachidonic acid and EPA, and the phycobiliprotein phycoerythrin [35]. Studies have demonstrated that the sulphated polysaccharides of *Porphyridium* sp. exhibit potential antiviral activity against herpes simplex virus (HSV-1 and 2) both in vitro and in vivo [55,56]. Furthermore, it has also been reported that different-molecular-weight subunits of its polysaccharides demonstrate important antioxidant and immunomodulatory activities [57,58]. Likewise, the Cyanobacteria *Phormidium* sp. is a recognized source of extracellular polymeric substances (EPS), which possess applications in the pharmaceutical, cosmetic, and food industries as an emulsifier and thickening agent [59]. Additionally, species of this genus have been reported to inhibit the growth of different Gram-positive and Gram-negative bacterial strains, yeasts, and fungi [60].

### Biochemical Composition

Microalgae produce a suite of biochemical molecules, and the cellular content of each fraction varies according to the specific strain of alga and their physiological responses to biotic and abiotic factors, e.g., light intensity, photoperiod, temperature, nutrients, and growth phase [61,62]. In fact, these factors not only affect photosynthesis and cell biomass productivity, but also influence the pattern, pathway, and activity of the cellular metabolism, which, consequently, modify the cell composition [63]. As such, due to their evolutionary and phylogenetic diversity, combined with the possibility of manipulating cultivation parameters to stimulate compounds’ biosynthesis, these microorganisms became extremely attractive for bioprospecting and potential exploitation as commercial sources of a wide range of biomolecules [1,64].

Bioactive compounds of microalgal origin can be sourced directly from primary metabolisms, such as proteins, fatty acids, and vitamins; or can be synthesized from secondary metabolism. Such compounds can present several biological activities, which might be used in the reduction and prevention of diseases (Figure 1). In most microalgae, bioactive compounds are accumulated in the biomass; however, in some cases, these metabolites are excreted into the medium, being known as exometabolites [2].

Chemically, microalgae compounds can be grouped into proteins/enzymes, lipids/fatty acids, carbohydrates, pigments, vitamins, minerals, and other compounds not included in these classes [65]. Even though the biochemical differences in microalgal classes and species are evident, protein is typically the major organic constituent (12–35%), normally followed by lipids (7–23%) and carbohydrates (5–23%). Yet, these proportions can drastically change under specific environmental conditions, as previously mentioned [63].

Proteins play an important role in the structure and metabolism of microalgal cells. They are a fundamental component of the membrane and light-harvesting complex, including numerous catalytic enzymes involved in photosynthesis [57]. The protein content of many species can compete, quantitatively and qualitatively, with conventional protein sources. In terms of quantity, several microalgae are reported to possess very high concentrations of protein, ranging from 42% to over 70% in certain Cyanobacteria, and up to 58% in *Chlorella vulgaris* on a dry weight basis. In terms of quality, microalgae contain all of the essential amino acids that mammals are unable to synthesize [61,66]. Moreover, some proteins, peptides, and amino acids also have biological functions associated with nutritional benefits and human health. Thus, these biopolymers can be used as nutraceuticals or included in functional food formulations [65].

Among the biochemical components, lipids have received the greatest attention regarding extraction and commercialization. When research on algal lipids first began, the major goal was aimed at biodiesel production. Nevertheless, the significant amount of polyunsaturated fatty acids (PUFAs) present in microalgae composition have provided considerably more commercial value to these bioactives as a nutraceutical and infant formulation supplement [61].

Microalgae lipid fraction is mainly composed of neutral and polar lipids, whose proportion varies along with the different growth phases, species, and environmental/culture conditions. Polar lipids possess a structural function, comprising the cell wall and organelle membranes, such as glycolipids and phospholipids [67]. On the other hand, neutral lipids are regarded as energy storage products, which include acylglycerols (mono-, di- and triglycerides), sterols, hydrocarbons, free fatty acids, and pigments [61,68]. The fatty acids in microalgae are biosynthesized through the addition of acetate (C-2) units; almost all are straight-chain and with an even number of carbon atoms, predominantly between C-12 and C-22. The main saturated fatty acids present in these structures are acids with 12, 14, 16, and 18 carbon atoms. A wide variety of unsaturated fatty acids are found in algae, with chains between 16 and 22 carbon atoms and double bonds in cis configuration [64].

Microalgae produce an interesting array of fatty acids, and they are reported to be the primary producers of some PUFAs in the biosphere, mainly omega (ω)-3 long-chain polyunsaturated fatty acids [69]. The importance of these compounds is based on the inability of humans to synthesize part of them; PUFAs play a key role in several bodily functions and processes, acting as a precursor of distinct biological molecules [70]. Examples of PUFAs produced by microalgae include the linolenic, eicosapentaenoic (EPA) and docosahexaenoic (DHA) ω-3 fatty acids; and the linoleic, gamma-linolenic (GLA) and arachidonic (ARA) ω-6 fatty acids [71].

Under optimal cultivation conditions, several species, especially those belonging to the genera *Botryococcus*, *Chlorella*, *Nannochloropsis*, *Neochloris*, *Nitzschia*, *Scenedesmus*, *Isochrysis*, *Dunaliella* and *Schizochytrium*, are described to show exceptionally high amounts of lipids in their cell mass. Regarding industrial applications, the eukaryotic microalgae *Chlorella vulgaris* (up to 58% dry weight), *Nannachloropsis oculata* (up to 69% dw), *Botryococcus braunii* (up to 75% dw), and *Scenedesmus obliquus* (up to 50% dw) have been reported to be promising lipid sources [72,73].

The colourful appearance of microalgae is derived from the presence of pigments, which absorb visible light and have a fundamental role in cell photosynthetic metabolism. The three major classes of these compounds are chlorophylls, carotenoids, and phycobiliproteins [74,75]. Chlorophyll *-a* is the primary pigment in all photosynthetic organisms; it absorbs most energy from the wavelengths of violet-blue and orange-red light, serving as a primary electron donor in the electron transport chain [76]. All microalgae contain one or more types of chlorophyll, which are classified according to their structural features and wavelength absorption [77]. The type *-a* is the only one found in Cyanobacteria and Rhodophyta, and the types *-a* and *-b* are found in Chlorophyta and Euglenophyta. Chlorophylls *-c*, *-d* and *-e* can be found in diverse marine microalgae and freshwater diatoms. The chlorophyll fraction usually represents about 0.5–1.5% of the cell dry weight [64].

Carotenoids are fat-soluble substances with colours varying from brown, red, orange, to yellow. These pigments perform two key roles in photosynthesis: the light absorption in regions of the visible spectrum and the photoprotection of the photosynthetic systems. All carotenoids directly involved in photosynthesis are called primary carotenoids, where they participate in the transferring of absorbed energy to chlorophylls; thus, expanding the light-absorbing spectrum of the cell. Primary carotenoids are structural and functional components of the cellular photosynthetic apparatus, making them essential for the survival of the cells [75].

Some microalgae species can also undergo a carotenogenesis process as a response to different environmental factors and culture stresses, e.g., high-intensity light, nutrient deprivation, and temperature changes [78]. These substances are categorized as secondary carotenoids, and they play a major role in cell protective mechanisms through the dissipation of most energetic states of chlorophyll, occasioned by excessive absorption of light [79,80]. The presence of these carotenoids hinders the formation of reactive oxygen species (ROS), providing these pigments with a significant antioxidant property. Examples of primary carotenoids are α-carotene, β-carotene, lutein, violaxanthin, zeaxanthin, and neoxanthin, whereas typical secondary carotenoids include astaxanthin, canthaxanthin, and echinenone [77,81].

In addition to chlorophyll and carotenoid, the pigment-protein complex phycobiliprotein is also commonly present in Cyanobacteria, Rhodophyta, and Cryptomonads. These complexes are deep-coloured water-soluble fluorescent cell components, which belong to the photosynthetic light-harvesting antenna [82]. According to their amino acid sequences and absorption spectrum, phycobiliproteins can be divided into four main classes, namely allophycocyanin (bluish-green), phycocyanin (blue), phycoerythrin (red), and phycoerythrocyanin (orange) [83,84]. The principal producers of microalgal pigments are the species *A. platensis*, *P. cruentum*, *H. pluvialis,* and *D. salina*, which are able to accumulate a significant amount of phycocyanin, phycoerythrin, astaxanthin, and β-carotene, respectively.

Equally representing a great fraction of the microalga cell, carbohydrates are the major products derived from photosynthesis and carbon fixation metabolism. These constituents are either accumulated in the plastids as reserve materials, e.g., starch; or become the main component of cell walls, such as cellulose, pectin, and sulphated polysaccharides [85,86]. A third possibility is the excretion of large amounts of polysaccharides into the extracellular medium, known as exopolysaccharides (EPS), supposedly in order to protect the cell from fluctuations in environmental conditions and/or predators [87].

The biomass carbohydrate content, similar to other microalgal compounds, also depends on the species and on the cultivation and environmental conditions. Green microalgae, for instance, synthesize amylopectin-like polysaccharides (starch) as reserve carbohydrates; Cyanobacteria synthesize glycogen (α-1,4 linked glucan), and red microalgae produce a polymer known as floridean starch (a hybrid of starch and glycogen) [88,89]. Moreover, a commonly found polysaccharide in a large number of species is chrysolaminarin, a linear polymer of β(1→3), and β(1→6) linked glucose units [89].

In addition to bio-macromolecules, microalgae constitute a valuable source of vitamins and minerals. Vitamin A, B_1_, B_2_, B_6_, B_12_, C, E, K, niacin, nicotinate, biotin, and folic acid are some of the examples found in these micro-organisms. In some microalgae genus, such as *Arthrospira*, *Chlorella*, and *Scenedesmus*, vitamin A, B_1_, B_2_, E, and niacin can achieve even higher levels than those found in vegetables [85]. Concerning the minerals’ content, it may represent around 2.2 to 4.8% of the total microalgae biomass dry weight, including calcium, phosphorous, magnesium, potassium, sodium, zinc, iron, copper, and sulphur [57]. Furthermore, several microalgae exhibit a high content of different phytochemicals compounds, such as polyphenols, alkaloids, among others [90,91].

## 3. Encapsulation

Encapsulation may be defined as a process in which a substance (active agent) is entrapped or coated by a carrier material, in order to form a particulate system. The encapsulated compounds are also designated as the core, fill, or internal phase, whereas the carrier substances can be identified as wall material, membrane, capsule, shell, matrix, or external phase [12]. This technique may be used to encapsulate compounds in the solid, liquid, or gaseous state in small particles, which can be classified as nanoparticles when the dimensions vary from 1 to 100 nm; or microparticles, when the dimensions range from 100 nm to 1000 µm [92,93].

The protective barrier provided by encapsulation offers several advantages. The primary reason to develop these particulate systems is to maintain the biological, functional, and physicochemical properties of the active agent [94]. The wall material serves as a protection from adverse environmental and processing conditions, such as the undesirable effect of light, temperature, moisture, and oxygen; therefore, contributing to an increase in stability and an extended shelf-life [95]. Another important advantage is the possibility of overcoming challenges that normally restrict the incorporation of certain substances into commercial products. Encapsulation allows the increase in solubility of a compound into a dissimilar medium, masking unpleasant flavours, enhancing the bioavailability and bioactivity, as well as controlling and targeting the release of the core material as a response to external conditions (pH, temperature, etc.) [15,94,96].

The retention of the core substance within a particle and its stability depends on several factors, comprising the desired physicochemical and functional properties of the final encapsulation system, along with the properties of the carrier material and the active agent. The major characteristics to be considered regarding the core and coating materials are their chemical nature, molecular weight, polarity, and solubility; while for the final particulate system, the entrapment efficiency, permeability, degradability, and release profile must be equally taken into account [97,98,99].

Encapsulation technology has been extensively researched and applied in diverse areas, such as the pharmaceutical, medical, food, cosmetics, chemical, and agricultural industries [98,100]. In the pharmaceutical field, for instance, encapsulation is a key strategy to assist specific drawbacks in the formulation development, as it is capable of promoting drug delivery to specific body sites, control drug release, act as diagnostic tools, and improve physicochemical properties, e.g., water solubility, which, in turn, can bring positive changes to the medical treatment, such as lowering the therapeutic dose and minimizing the side effects [101,102,103,104]. Similar advantages can be related to the application of encapsulation in the cosmetics segment, where particulate systems could lead to a sustained release of the active agent, as well as an enhanced and deeper skin penetration [105,106].

Likewise, the food industry can particularly benefit from the use of encapsulation, especially regarding the development of functional foods. The addition of biologically active compounds into food has emerged as an exciting health-promoting strategy in recent decades; however, it may present several limiting factors, including high sensitivity to processing conditions, short shelf-life, fast-release of flavour during storage, limited uptake and bioavailability, lack of compatibility and uniformity with the food matrix, or degradability through the gastrointestinal tract passage [16,107,108]. In light of this, encapsulation represents a useful tool for the suppression of the aforementioned limitations, since it enables the protection of a wide range of compounds by their entrapment into a protective matrix [109].

### 3.1. Structure and Composition

The structure of an encapsulation system depends upon the arrangement of the core substance and deposition process of the coating material, which can be broadly divided into reservoir or matrix systems [16]. The reservoir systems are also referred to as nano- or microcapsules and can be further classified into mono-core, multi-core, or multi-shell mono core; where the first contains an outer shell around a single core, the second has distinct cores entrapped into a shell, and the last comprises a single core surrounded by many shells. On the other hand, the matrix system is when the active agent is uniformly distributed within the coating material network. The particulate systems are normally formed as spherical shapes; however, non-spherical/irregular configurations can be found for all the systems previously described [15,110]. A resume of all these structures is demonstrated in Figure 2.

Regarding the composition of an encapsulation system, the encapsulated active agents can have a hydrophilic or lipophilic nature, comprising several classes, such as drugs, vitamins, minerals, nutraceuticals, antimicrobials, antioxidants, flavours, enzymes, essential oils, colourants, among others [93]. Additionally, the coating material can be selected from a wide variety of natural and synthetic polymers, co-polymers, and bio-based substances. This choice will depend on the active agent to be encapsulated and the properties desired for the final system [112]. Important aspects to be considered are mostly the solubility, stability, release properties, and safety; thus, for use within the food industry, the substance must have a food-grade status; and for pharmaceutical applications, it should present, among other criteria, biocompatibility and biodegradability [113]. A resume of the most used coating materials for encapsulation purposes can be found in Appendix A.

Materials derived from natural sources may be classified as (i) carbohydrates, such as starch, maltodextrin, pectin, cellulose, cyclodextrin, and inulin; (ii) proteins, such as gelatine, whey protein, casein, bovine serum albumin, and different vegetable sources; (iii) waxes or fats, including glycerides and phospholipids, or (iv) gums, such as Arabic, guar, and mesquite. In addition to these, the polysaccharides chitosan and alginate are mostly investigated for food and pharmaceutical applications [16,112,114]. The main advantages of using these bio-based materials are that they are normally abundant in nature, biodegradable, and biocompatible. On the other hand, as they represent natural resources, they can display unstable properties due to batch to batch variation [15].

Another range of coating materials used for the encapsulation process is synthetic polymers, which can be further classified into biodegradable and non-biodegradable. Among the biodegradables ones, aliphatic polyesters, such as poly (ε-caprolactone) (PCL), poly (lactic acid) (PLA), and poly (lactic-co-glycolic acid) (PLGA) are frequently explored in the pharmaceutical industry [115]. Other examples of this group also comprise the polyanhydrides, polyamides, polyurethanes, and phosphorus-based polymers. Concerning the non-biodegradable polymers, cellulose derivatives, such as carboxymethyl cellulose, ethyl cellulose, or cellulose acetate, are also broadly applied, along with poly(ethylene glycol) (PEG), polyvinyl alcohol (PVA), and poly(N-vinylpyrrolidone) (PVP) [113,116]. Differing from natural materials, synthetic polymers exhibit a higher chemical and mechanical stability, with the possibility to modify their properties according to the desired final system. Nevertheless, low biocompatibility and biodegradability may represent the main drawbacks [15].

Furthermore, wall materials can also suffer a functionalization process, which offers the possibility to obtain encapsulation systems with modified properties, different from those normally found in the literature, e.g., an increase in biodistribution and additional bio-marker function [117,118].

### 3.2. Encapsulation Techniques

Several techniques have been proposed in the literature for encapsulation processes; nevertheless, there is not a specific method that could be regarded as a standard and suitable for the different active agents and encapsulated systems. Before choosing a technique, the crucial aspects to be analysed are the type and properties of the core and coating materials, and the proposed application and characteristics of the final system [12,97,119]. Moreover, the size, shape, and internal structure of the particles vary considerably depending on the selected production method [120]. Some of the most described encapsulation techniques are spray-drying, spray-cooling/chilling, fluidized bed, coacervation, solvent evaporation, liposomes, supercritical fluid technology, interfacial polymerization, nanoprecipitation, emulsification, inclusion complexation, electrospray system, and extrusion [94,95,96,117,119]. It is noteworthy mentioning that for a widespread application, the chosen method should be cost-effective, easy to scale-up, and have its safety status considered [99].

The strategies involved in the synthesis and production of encapsulation systems are based on two main approaches, namely bottom-up and top-down. The bottom-up approach is defined when large structures are built or grown through atom-by-atom or molecule-by-molecule techniques, mediated by different interactions, e.g., van der Waals, ionic and hydrophobic interactions [121]. This approach includes chemical synthesis, self-assembly, and positional assembly of molecules, which are influenced by several physicochemical and environmental factors, such as pH, temperature, concentration, and ionic strength [121,122]. Examples of techniques that follow this approach are nanoprecipitation, coacervation, inclusion complexion, and supercritical fluid encapsulation [123].

Conversely, the top-down approach involves physical processing of the core and coating materials, which requires precise tools focusing on the size reduction and shaping of the structure for the desired application. Extrusion, homogenization, electrospinning/spraying, and emulsification-solvent evaporation are examples of top-down techniques [120,122,123]. In general, the bottom-up techniques are considered more advantageous than the top-down approach, as they allow greater control over the properties of the particles, namely the size, morphology, and physical state, and they are also less energy-consuming. Moreover, the risk of sample contamination is often significantly reduced compared to top-down technologies [99,124]. A schematic representation of the main reported encapsulation techniques of both approaches is shown in Figure 3.

Spray drying is one of the oldest and the most widely used encapsulation techniques, mainly in the food sector. Firstly, the active agent is dispersed or dissolved in an aqueous solution or one prepared with the coating material. The mixture is then atomized, where little droplets are formed and dried through hot circulating air [109]. The size of the particles normally varies from 1 to 50 µm, but it can be reduced to 0.2 µm by using a nano spray dryer. The encapsulation efficiency is influenced by different process parameters, including the viscosity and surface tension of the solution, the solubility of the core, or even the air entrance temperature, air flux, and humidity [125]. The main advantages of this technique are its simplicity, flexibility, fast production, and low operating costs. On the other hand, the particles formed may not be uniform and some sensitive compounds could be degraded by the high air temperatures [92].

The extrusion technique is another quite common choice to obtain an encapsulation system. It consists in the passage of a solution composed of the coating material and active agent through a nozzle, reaching a gelling environment. Several methods have been used to form extruded particles, including electrostatic extrusion, simple dripping, vibrating jet/nozzle, and melt extrusion. Following the particle formation, they must be instantly hardened to capsules by either physical processes, e.g., cooling or heating, or chemical processes, e.g., gelation [110]. Extrusion is a simple and low-cost technique, which is suitable for labile substances when only a final gelation process is required, yet it may present low encapsulation efficiency [126].

Coacervation consists of the formation of two immiscible phases from a solution containing a dispersed polymer. The substance to be encapsulated is dispersed in a polymeric solution, which will act as the coating material. Through different methods, the polymer separation is induced, creating a new phase (coacervate) [127]. The particle formed can be collected by centrifugation or filtration, followed by washing, drying, or hardening. This technique can be further divided into complex or simple coacervation: the simple type is promoted by a change in the medium, which causes a desolvation in the coating material; while in the complex type, there is a mutual neutralization of two polymers with opposite charges that will compose the coating [128]. Among the factors that affect the particle size, which can vary from 20 to 200 µm, are the stirring rate, the phases’ viscosity, the type and concentration of the surfactant (if added), and the temperature. It is possible to achieve high entrapment efficiency and good control of the particle size through coacervation. However, the drawbacks reported for this technique are particle agglomeration, as well as the high operational cost [129].

Emulsification is a technique continuously applied in the food, pharmaceutical, and cosmetic industries. Briefly, an emulsion consists of a system formed by at least two immiscible liquids, generally water and oil, where one of the liquids is dispersed as small spherical droplets in the other, surrounded by a thin interfacial layer of surfactant molecules. The systems where oil droplets are dispersed in an aqueous phase are called oil-in-water emulsion (O/W), whereas the systems where water droplets are dispersed in an oily phase are called water-in-oil emulsion (W/O). The addition of surfactants in the emulsion system is frequently necessary to obtain a kinetically stable solution. The emulsion particle diameter normally varies from 0.1 to 100 µm [94]. Emulsions are prepared through the homogenization of the water and oily phases, together with one or more surfactants, using different methods, such as high-pressure homogenization, microchannel emulsification, membrane emulsification and ultrasound, among others. The main advantages of this technique are the relatively easy preparation and low cost; nevertheless, emulsions may suffer from physical instability when exposed to diverse storage and processing conditions, which could lead to additional processing steps or the incorporation of additives to improve stability [130]. Another technique based on the emulsification process is the emulsification-solvent evaporation, which consists of forming an emulsion of a polymer solution (coating material) and an active agent in a volatile organic solvent, followed by evaporation of the solvent, usually under atmospheric conditions [131]. This technique is a simple method to obtain small droplets with a narrow size distribution. However, the high amounts of organic solvent used may increase production costs [132].

Considering pharmaceutical applications, liposomes are one of the most researched encapsulation processes. Such a technique involves the formation of lipid vesicles from aqueous dispersions of amphiphilic molecules, e.g., polar lipids, which tend to produce bilayer structures. Liposomes are typically spherical, with sizes varying from nanometre to micrometre range. The vesicles formed may contain a single or multiple layers of amphiphilic polymolecular membranes [133,134]. The possibility of encapsulating both lipophilic and hydrophilic compounds, promoting targeted drug delivery, and its versatility in terms of size and number of layers comprise its main advantages. On the other hand, the potential toxicity due to organic solvent residues, high cost in large scales, and low stability are considered the main processing obstacles [135].

## 4. Microalgae Encapsulation

### 4.1. Functional Foods

Functional food, in general terms, may be defined as a natural or processed food, which contains an identified component, in qualitative and quantitative amounts, with a proven and documented health benefit [136,137]. This concept was created in recent decades, opening a new research field that is in constant expansion due to consumers’ increasing awareness of the close correlation between diet and health. Beyond providing nutrients required for the bodily metabolism, it is well-known that food may play a key role in the prevention and treatment of certain diseases, along with the improvement of physical and mental well-being [138,139]. Following this trend, safety issues regarding the consumption of processed foods have also become a concern. National authorities, such as the Food and Drug Administration (FDA) and the European Food Safety Authority (EFSA), have restricted the use of many synthetic additives in food, e.g., synthetic dyes, due to a growth in cancer development or allergic reactions [6,51].

Accordingly, there is a great interest in the investigation of natural resources and biologically active compounds with high nutritional value and functionality to be used as a food ingredient in the development of novel functional foods [140]. Among these, microalgae are emerging as a valuable and economically viable alternative, as they represent a rich source of food-grade compounds and almost an unlimited field of exploration due to their abundant taxonomic diversity [140,141]. A variety of microalgae biomass has already been successfully applied in the fortification of assorted food products, such as cookies, bread, pasta, and some dairy goods [142].

On the other hand, the incorporation of nutraceutical compounds into food is more of a challenging approach. The effectiveness of a bioactive as a health-promoting substance within the food matrix depends on keeping its functionality intact during food processing and storage; conserving the characteristics (taste, texture, colour, smell, etc.) and acceptability of the original food; and lastly, assuring the bioavailability of the active ingredients, which includes sustaining sufficient time of gastric residence without degradation and appropriate gut permeability [95]. Due to the inherent instability of most bioactive compounds present in microalgae, the efficacy of this process may be compromised. Consequently, their incorporation into encapsulation systems seems to be a promising strategy to deliver microalgae health benefits at boosted levels through functional foods [96].

One of the most explored microalgae concerning encapsulation systems for food applications is the species *Haematococcus pluvialis.* Several research groups have investigated the encapsulation of its extract obtained by different methods or purified compounds, essentially the carotenoid astaxanthin. A resume of the systems reported in the literature and their main findings are described in Table 1.

Astaxanthin, which possesses a singular and recognized antioxidant potential, is present in a considerable amount in the *H. pluvialis* cyst cells formed under adverse environmental conditions. Nevertheless, the pure addition of the whole cells into food may not be applicable; during the cyst phase, the bioactive compounds are surrounded by a thick cell wall, which could hinder proper release and bioavailability. Therefore, an extraction method is normally required to achieve cell wall disruption and obtain the compounds of interest [170]. When the extraction process is concluded, highly sensitive substances, such as astaxanthin, once protected by the cell wall, are now susceptible to the effects of light, oxygen and high temperatures, among others, which explains the high number of studies applying encapsulation for this compound [171].

In Table 1, it is possible to observe that all the studies have used as core substance the already disrupted *H. pluvialis* cell; purified astaxanthin, as powder or oleoresin; or an astaxanthin-rich extract, always obtained through green techniques. The coating materials selected for those encapsulation systems were majoritarian natural polymers, mainly chitosan, alginate, whey protein, maltodextrin, and Arabic gum, as they are food-grade and widely used as food additives. Regarding the encapsulation techniques applied for *H. pluvialis* and its compounds, the diversity reported in the studies is clear, including both top-down and bottom-up approaches. Among them, spray drying, extrusion, and emulsification were preferred by half of the authors for the encapsulation processes.

Despite all the differences found for astaxanthin encapsulation, the protective potential of this strategy over such a compound is unquestionable. The coating layer was reported to promote stability improvement under storage at adverse conditions, such as high temperatures, oxygen exposure, or extreme pH values, while preserving its antioxidant potential. Nonetheless, greater stability was usually detected when the encapsulation system was stored at low temperatures. Moreover, the enhancement in astaxanthin bioaccessibility and bioavailability was also investigated, obtaining positive results in in vitro studies.

The development of an encapsulation system is a complex process; as such, understanding the influence of the type and concentration of the coating material or the applied technique, on the particle physicochemical properties is also a significant step to guarantee its functionality when added into a food matrix. Anarjan and Tan [167], for instance, investigated the emulsification and stabilization ability of four different polysaccharides, namely Arabic gum, xanthan gum, pectin, and methylcellulose, in the preparation of water-dispersible astaxanthin nanoparticles. The authors reported that the physicochemical characteristics of the prepared nanodispersions were significantly influenced by the type and chemical structures of the polysaccharide used as the coating material, with the one produced with Arabic gum showing the smallest average particle diameter and highest physical stability. However, they observed a considerable astaxanthin degradation after 30 days of storage for all samples, which allowed them to conclude that, generally, the nanodispersions produced with polysaccharides have a larger average particle size and less physicochemical stability than those obtained with proteins and small molecule emulsifiers.

The encapsulation of bioactives from the marine microalga *Phaeodactylum tricornutum* for food applications was described in two different studies, both using the electrospray/electrospinning technique. This species is considered a significant source of the carotenoid fucoxanthin and PUFAs, which have been associated with several health-promoting properties. Nevertheless, the rough algal extract is not suitable for food fortification due to its distinctive odour, consistency, and low bioactive concentration. Additionally, such lipophilic bioactives possess an inherent sensitivity to many adverse environmental conditions, low water solubility, compromised bioavailability, and potential degradation during digestion.

Seeking to overcome these issues, Koo et al. [172] researched the development of a fucoxanthin-enriched fraction from *P. tricornutum*-loaded nanoparticles to improve the application of this carotenoid into the food industry. The nanoparticles were prepared through homogenization followed by an electrospray system, firstly using only casein as the coating material, then followed by an extra layer of chitosan. In vitro simulated digestion studies have demonstrated a better bioaccessibility of the nanoparticles over the *P. tricornutum* powder. Such a result was also corroborated by the in vivo pharmacokinetic assay, where the casein-chitosan nanoparticles exhibited superior bioavailability, possibly due to increased retention or adsorption to the mucin by the presence of chitosan. In another study, Papadaki et al. [51] recovered a lipid fraction from *P. tricornutum* through ultrasound-assisted extraction using coconut oil as a solvent. Subsequently, the extract was emulsified and encapsulated in ulvan:pullulan nanofibers by electrospinning. The encapsulation process showed an entrapment efficiency of 90%, for both carotenoids and PUFAs, in food-grade water-based polysaccharides; thus, representing a promising strategy for incorporation of lipophilic bioactives from the microalga *P. tricornutum* into food matrices.

The cultivation and bioactive extraction optimization of the microalga *Dunaliella salina* are topics constantly researched in the literature. As the richest natural source of the carotenoid β-carotene, the encapsulation of this species is also a trending area when it concerns functional foods. Techniques that were already investigated comprise calcium alginate beads followed by fluidized bed drying [173] and spray-drying using a mixture of maltodextrin:Arabic gum [174] or different combinations of gelatine, maltodextrin and Arabic gum, as coating materials [175]. All the researchers concluded that encapsulation was able to promote stability improvement in the β-carotene content naturally present in *D. salina*; however, they also reinforced that better results can be achieved through the utilization of lower temperatures in the drying process, with the absence of light and high temperatures during storage.

The encapsulation of microalgae of the genus *Chlorella* has been widely investigated for environmental monitoring, but its use in the food industry has not been fully explored yet. Differing from what has been published about other microalgae species, *Chlorella* was considered as a possible coating material in the encapsulation system of other bioactives. *Chlorella vulgaris* cells were investigated as a carrier for the polyphenol curcumin [176] and *C. pyrenoidosa* cells as a carrier for the carotenoid lycopene [177], aiming at protecting the core substance while developing an innovative nutraceutical complex. The encapsulation process in both studies was performed by adsorption. Results demonstrated an increase in the photostability of curcumin by about 2.5-fold, and an improvement in the thermal and storage stability of lycopene when loaded into *Chlorella* cells. Moreover, the *Chlorella*–lycopene complex presented higher antioxidant activity when compared to the same amount of free lycopene at room temperature for 25 days, which might be partly due to the carrier protection, and partly due to the endogenous antioxidants present in *C. pyrenoidosa* cells.

The species *Chlorella pyrenoidosa* was also chosen as the object of study of Wang and Zhang [178], where they evaluated the extraction and antitumor activity of a polypeptide obtained from this microalga to further encapsulate through two different techniques, namely complex coacervation and ionotropic gelation. The antitumor activity was confirmed to have inhibitory activity on human liver cancer HepG2 cells and encapsulation was carried out as a solution to avoid stomach degradation, followed by a proper release in the intestinal environment. The in vitro release assay revealed that the encapsulated *C. pyrenoidosa* polypeptide was well preserved against gastric enzymatic degradation, increasing its bioavailability at least two-fold when compared to the non-encapsulated bioactive.

Another microalga that has been investigated for food purposes is the species *Phormidium valderianum.* Chatterjee et al. [179] reported the encapsulation process by spray-drying of an antioxidant-rich fraction of *P. valderianum* obtained through supercritical carbon dioxide extraction, aiming at enhancing the storage stability of the extracted compounds. A mixture of maltodextrin:Arabic gum was selected as wall material and the optimization of the microencapsulation process parameters was performed to achieve the best yield and biological properties, which were examined by antioxidant capacity, phenolic content and reducing power. The condition that provided the best response combination of the analysed parameters was spray-drying at an inlet temperature of 130 °C, with wall material composition of maltodextrin:Arabic gum (70:30). Additionally, a stability study was also carried out for 60 days, comparing the IC_50_ values of the DPPH (2,2-diphenyl-1-picrylhydrazyl) antioxidant assay of non-encapsulated and encapsulated microalgal extract. As a result, it was confirmed that the encapsulation process was able to protect the antioxidant compounds for a longer period, enhancing the antioxidant activity shelf-life by eight-fold.

Similarly, Bonilla-Ahumada et al. [180] investigated the microencapsulation of fresh biomass from the microalga *Tetraselmi chuii* by spray-drying, along with the effect of the wall material (maltodextrin:Arabic gum (60:40), chitosan 3% or gelatine 2%) and processing conditions (inlet temperature 110, 130, and 150 °C) on the preservation of β-carotene and other antioxidant compounds present in this species. The work reported preservation of 80–92% of β-carotene and 46–81% of the phenolic compounds in freshly microencapsulated microalga, even after three months of storage in the dark,at 25 °C, when coated with maltodextrin and spray-dried at 130 °C. Moreover, the authors emphasized the advantage of using spray-drying regarding algal biomass, as it is capable of protecting unstable metabolites, as well as facilitating the transport and further incorporation into food products.

The encapsulation process is also widely employed for microalgae of the genus *Arthrospira*, focusing on improving several challenges involved in the incorporation of its biomass/powder, extracts, or compounds into functional foods. A compilation of published studies can be found in Table 2. The species *A. platensis*, the main representative of this group, is acknowledged as a pronounced protein source and rich in many essential nutrients for the human diet. The fortification of different food products with whole *A. platensis* biomass has already been explored by many authors seeking to increase their nutritional content and functionality, i.e., antioxidant potential [181,182,183]. However, encapsulation may provide not only a protective layer for stability enhancement over processing and storage conditions, but the possibility to achieve more uniform distribution in the food matrix.

The addition of microencapsulated *A. platensis* powder obtained through spray drying into yoghurt was investigated by Da Silva et al. [189] and compared to a formulation containing the free microalga. The authors reported that microencapsulation was able to promote higher thermal stability, showing better anti-inflammatory activity without exerting cytotoxicity. Moreover, the yoghurts incorporated with encapsulated *A. platensis* exhibited a more homogeneous appearance, lighter green colour, and noticeable decrease in the strong odour, whilst, at the same time, maintaining yoghurt’s nutritional profile and an improved antioxidant activity throughout the storage time. Recently, Zen et al. [192] developed a functional pasta fortified with *A. platensis* biomass-loaded alginate microparticles also through spray-drying. Even though the pasta properties were affected by the addition of microparticles, the overall acceptability index was not influenced according to sensorial studies. Most importantly, microencapsulation was able to protect 37.8% of the biomass antioxidant potential from the pasta cooking conditions.

On the other hand, the addition of *A. platensis* extracts and the protein-pigment phycocyanin—its main antioxidant compound—into food suffers from the limitations previously described for food fortification with natural bioactives. The extract composition obtained from this microalga is highly dependent on the extraction technique and, mostly, on the type of solvent used. Aqueous-based extracts are essentially rich in phycocyanin, phenolic compounds, and other polar substances, while organic-based extracts are rich in chlorophyll, carotenoids, and other lipophilic compounds [197].

Among all the studies, the encapsulation of isolated phycocyanin was investigated by a considerable number of authors using different techniques, such as spray drying, extrusion, and electrospraying. The particles’ properties were analysed and optimized to achieve the best coating material concentration or composition of two distinct types, with the highest entrapment efficiency, particle size consistency, and stability. Some authors also explored the stability of encapsulated phycocyanin, describing thermal stability improvement and resistance to the acidic environment when alginate and chitosan were used together as coating materials [184] and thermal resistance up to 216 °C with full preservation of its antioxidant activity when encapsulated with PVA [194]. Concerning the encapsulation of *A. platensis* extract, phenolic-rich, carotenoid-rich, aqueous-based, and a commercial powder extract were evaluated as the core of four different encapsulation systems. The beneficial effects of encapsulation were confirmed through different outcomes, comprising gastric protection of the phenolic extracts, high stability of the carotenoids, antioxidant potential over storage, and preservation of colour stability and antioxidant potential of the encapsulated aqueous-based extract.

### 4.2. Pharmaceutical

Naturally derived products have served as a vital source of drugs since ancient times. Nowadays, approximately one-third of the top-selling pharmaceuticals are of natural origins or their derivatives. Plants and microorganisms represent a practically unlimited source of biochemical molecules, which may have promising pharmacological activities and therapeutic benefits in the treatment of diverse diseases [198]. In particular, microalgae have shown their importance in the discovery of new therapeutic molecules, as well as in the isolation and characterization of already acknowledged ones [199].

As previously mentioned, the application of natural compounds in therapeutics faces significant shortcomings and developmental challenges, highlighting their usually poor aqueous solubility, inherent instability, and low bioavailability [200]. The use of micro/nanoencapsulation has been shown as a solution by the pharmaceutical industry to address the issues associated with these drawbacks, where the therapeutic value of biologically active compounds can be drastically improved [198]. Microalgae, as a rich and valuable universe of natural products with proven pharmacological properties, are assumed to benefit from this strategy. However, the application of these microorganisms in drug delivery systems for pharmaceutical purposes is still a field to be explored.

Similar to what has been reported for microalgae encapsulation in food applications, the species *H. pluvialis*, particularly its main bioactive compound astaxanthin, is the most researched one regarding disease treatment. There is a vast number of biological activities associated with this carotenoid; however, studies involving encapsulation strategies only consist of a few examples (Table 3). As can be observed in Table 3, liposomes and nanoemulsion were the encapsulation techniques chosen by most of the authors, aiming to improve the biopharmaceutical properties of astaxanthin. Through the results, it was possible to confirm that most of the astaxanthin’s biological activities are due to its unique antioxidant potential, which is able to protect against diverse deleterious effects of oxidative stress.

The use of encapsulation, nonetheless, significantly boosted the outcomes. The anti-aging activity of astaxanthin-rich extract loaded nanofibers was investigated by Nootem et al. [201] and not only was a strong potential against oxidative stress reported, but the nanofibers also promoted a slower in vitro release profile and increased the stability of the core compounds in comparison with the free extract. In another study, Chiu et al. [202] proposed that astaxanthin-loaded liposomes could be beneficial to lipopolysaccharide (LPS)-induced acute hepatotoxicity, which is expressively related to oxidative stress. The results indicate that, in fact, encapsulated astaxanthin had its bioavailability and liver cell uptake enhanced, and that the developed drug delivery demonstrated in-vivo hepatoprotective and acute anti-inflammatory effects, with even superior results than the one found for the positive control N-acetylcysteine. Overall, astaxanthin therapeutic properties may profit deeply from drug delivery systems, presenting enhanced effects without cytotoxicity when compared to the free molecule.

Likewise, bioactive compounds from microalgae of the genus *Arthrospira* were also considered as the core substance of encapsulation systems focusing on pharmaceutical applications. The protein C-phycocyanin was the compound with a major interest in this regard; however, phenolic and free fatty acid-rich extracts were equally investigated. A resume of a literature review comprising the encapsulation of *Arthrospira* bioactives as drug delivery systems can be found in Table 4. According to these studies, the encapsulation of these microalgae compounds for skin delivery has been particularly explored. Phycocyanin is correlated with several biological properties, whose therapeutic applications may be challenged by its molecular features (instability and high molecular weight) and the gastrointestinal acidic environment. Aiming to overcome these issues, Hardiningtyas et al. [203] studied the possible transdermal permeation of phycocyanin in a solid-in-oil nanodispersion, which was successfully achieved; the developed encapsulation system was able to facilitate the accumulation of phycocyanin in the *stratum corneum*, followed by its permeation into deeper skin layers.

Concerning the treatment of cutaneous diseases, the anti-inflammatory potential of phycocyanin-loaded liposomes [213] and the anti-biofilm growth activity of *A. platensis* fatty acid-loaded coper-alginate nanocarriers [214] were investigated, exhibiting positive effects due to the combination of the bioactives and encapsulation systems. In the first work, liposomes improved phycocyanin accumulation in the whole skin, as well as the anti-inflammatory response, which was confirmed by its superior results when compared to the free phycocyanin gel; while in the second study, the encapsulated fatty acids were able to inhibit half of the film formation with a very low dosage in 24 h.

The process of encapsulating an active compound also represents a possibility of enhancing its time of residence in the body; thus, increasing the cell uptake; or promoting an active targeting, which could be achieved through particle surface functionalization. In this context, Yang et al. [216] hypothesized that polysaccharides from *A. platensis* could be used as a surface decorator of nanomaterials to prevent plasma protein adsorption, maximize circulation time, and improve their cell-penetrating abilities, more specifically cancer-targeting ability. Selenium nanoparticles were then prepared and functionalized with purified *A. platensis* polysaccharides and its cell uptake, cytotoxicity, and in vitro anticancer activity were evaluated. Results show that the polysaccharides’ surface significantly enhanced the cell-penetrating and apoptosis-inducing abilities of the selenium nanoparticles towards several human cell lines; especially in A375 human melanoma cells, where they were found to be extremely susceptible to the functionalized particle (IC_50_ of 7.94 µM). Accordingly, *A. platensis* polysaccharides were suggested as a potential enhancer of the anticancer activity of nanomaterials.

Equally focusing on anticancer targeting, the microalgae *Chlorella protothecoides* and *Nannochloropsis oculata* had their lipid extract incorporated into two different encapsulation systems in the study performed by Karakas et al. [217], aiming at bioactive protection and assessment of the in vitro cytotoxicity activity in human cancer cells. The nano-microparticles were obtained by electrospray and microemulsification techniques, using PVA:chitosan or PVA:sodium alginate, and calcium alginate, respectively. Based on the MTT test (3-(4,5-dimethylthiazol-2-yl)-2,5-diphenyltetrazolium bromide), it was possible to confirm that the encapsulated microalgae extract exhibited cytotoxicity in the cancer cell lines from brain glioblastoma and colon colorectal carcinoma, while no effect was observed in healthy cells.

Finally, the encapsulation of *Dunaliella salina* extract was carried out by Zamani et al. [218], seeking to develop an oral drug delivery system for gastric protection and release control of the microalga compounds. Arabic gum-coated magnetic nanoparticles were selected as the encapsulation system for the extracts obtained at the logarithmic and stationary growth phases; and their properties, such as release profile, antioxidant and anticancer capacity, were assessed. The authors reported that both formulations promoted a sustained release of *D. salina* extract in PBS at pH 4.5 and 7.2, with final relative release values of 72.41 and 43.51% for logarithmic and stationary phases over 48 h, respectively. Moreover, the antioxidant and cytotoxic activity of the free and nanoparticulated extract on MCF-7 and HeLa cells indicated that both phases presented strong antioxidant and anticancer effects in a time and dose-dependent manner. Therefore, it was concluded that the oral delivery of encapsulated *D. salina* extract seems to be an effective approach to reduce adverse gastric effects and maintain the functionality of its compounds.

### 4.3. Cosmetics

Cosmetics are a class of products aimed at improving the structure, morphology, and appearance of skin or external parts of the body. A large section of this segment comprises skin topical formulations, which are composed of excipients and one or more active ingredients. Following the current global trend for products derived from natural sources, there is a demand for the development of environmentally sustainable cosmetic products, with less chemical compounds, which could act as cosmeceuticals [219].

The interest in microalgae regarding cosmetics application is relatively recent; these microorganisms produce metabolites in response to changes in the environment, whose main function is linked to the cell’s ability to regenerate and self-protect against external adverse conditions. In this context, it is assumed these compounds could instigate the equivalent effect when applied on the skin. Among the bioactives extracted from microalgae that can be potentially used in cosmetics formulation are the ones with pronounced antioxidant activity, such as astaxanthin and C-phycocyanin [220,221].

The skin is the outer organ of the body and therefore acts as the primary barrier against the loss of endogenous substances, as well as the penetration of external agents into the human body. As it constitutes an interface with the environment, the skin is considered a target of several exogenous factors, such as UV radiation, pathogens, pollution, and other toxic compounds. Such factors are usually associated with excessive production of reactive oxygen species and other free radicals, which are pro-inflammatory mediators and may induce many deleterious effects, including DNA damage, oxidative stress, photoaging, and carcinogenesis [221,222]. As such, microalgae bioactives could play an advantageous role in maintaining the skin health status and in the treatment of some dermatological issues, such as hyperpigmentation, dehydration, photo-oxidation, photoaging, as well as protection against skin cancer [219,223].

However, the topical application of natural bioactives may be limited not only by their chemical instability in terms of product development, but by their poor water solubility, which might restrain the skin absorption and lead to low bioavailability. Additionally, some bioactives also possess a high molecular weight, which makes their permeation through the first skin layer, the *stratum corneum*, impracticable [224,225]. Given these conditions, it is imperative to develop an appropriate and efficient delivery system for microalgae bioactives onto the skin, which have already been described by a few authors. The encapsulation of *H. pluvialis* in liposomes was performed by Hama et al. [226], and its protective effect on ultraviolet-induced skin damage through topical application was investigated. Firstly, the authors analysed and confirmed the powerful in vitro antioxidant activity of astaxanthin-loaded liposomes, which was followed by an in vivo assay. The topical application of the developed encapsulation system was then capable of inhibiting UV-induced skin damage, collagen degradation, and melanin production; hence, showing its potential as a protective formulation against UV-induced skin disorders.

In another study, Sun et al. [224] developed an astaxanthin non-aqueous nanoemulsion through a high-pressure homogenization method for topical application, to combine the advantages of an encapsulation system and non-aqueous emulsions. Results show that the system was able to avoid astaxanthin degradation, keeping its stability for over 4 weeks at 25 °C. Additionally, when compared to traditional water-based emulsions, the non-aqueous type could effectively improve astaxanthin chemical stability against light and high temperatures. In vitro cell assays revealed that the non-aqueous nanoemulsion had low toxicity and protected the cells against oxidative stress. Moreover, in vitro permeation studies and skin section histological analyses exhibited the enhanced permeation of astaxanthin with low systemic absorption and unchanged epidermis, which proved the efficacy and safety of the astaxanthin-loaded non-aqueous nanoemulsion for topical application of that carotenoid. Likewise, Hu et al. [223] prepared and optimized astaxanthin-loaded PLGA nanoparticles through the emulsification-solvent evaporation technique and investigated its cellular uptake, cytotoxicity, and photodamage protective effect in human keratinocyte (HaCaT) cells. According to the in vitro study, the optimized nanoparticles exhibited excellent cell viability and cell uptake, as well as low cytotoxicity. Additionally, the photodamage assay demonstrated that the nanoparticles presented higher antioxidant activity compared to pure astaxanthin after exposure to UVB radiation and were able to resist photodamage in the cells by reducing ROS levels and restoring mitochondrial membrane potential.

The encapsulation of the microalga *Arthrospira sp*. for cosmetic purposes was interestingly explored by Byeon et al. [227]. The study aimed at the development of an *Arthrospira* sp. extract-impregnated nanofiber patch in a double-layer form, which was supported by a PCL nanofibrous cover matrix; both prepared through the electrospinning technique. The mechanical stability, cytotoxicity, water absorption, and extract release profile were assessed by the authors. As result, the patch was found to be non-cytotoxic in human cell-based tests and it presented more moisture and better adhesiveness than the patch prepared with only alginate nanofibers, which indicates the *Arthrospira sp*. extract enhanced those properties, in addition to its biological effects on the skin. Furthermore, the dry patch promoted the release of most of the extract onto the skin within 30 min, suggesting its potential to be an innovative skincare product.

The bioactive phycocyanin for food and pharmaceutical applications reported in this review was commonly derived from the microalga *Arthrospira* sp. However, the species *Aphanizomenon flos-aquae* is also a rich source of this compound and was the object of study of Castangia et al. [225] for cosmetic purposes. The authors encapsulated phycocyanin in hyalurosomes, a type of phospholipid vesicle immobilized with hyaluronan sodium, or alternatively in PEG hyalurosomes, due to the high molecular weight and consequent low bioavailability of this compound. The skin delivery and the protective potential against oxidative stress damage of these encapsulation systems were assessed through in vitro cell-based permeability, biocompatibility, and antioxidant activity assays. The permeation studies demonstrated that hyalurosomes favoured phycocyanin deposition in the deeper skin layers, mainly when the permeation promoter PEG was added to the particle surface. Results also show that the phycocyanin-loaded hyalurosomes were highly biocompatible, with improved phycocyanin antioxidant activity on stressed human keratinocytes when compared to the free compound, also promoting control in inflammation and a stimulus in keratinocyte proliferation.

## 5. Conclusions

The interest in microalgae for industrial applications has been growing in the last decade due to the vast collection of high value biologically active compounds produced by this group. These bioactives are associated with several pharmacological properties, which have been demonstrated to promote beneficial effects for human health. Additionally, consumers’ awareness towards the use of healthier, sustainable, and safer ingredients stimulate the exploration of new natural resources to be applied in the food, pharmaceutical, and cosmetic industries.

Considering the physicochemical limitations and technological challenges reported for the incorporation of bioactives into products, namely high instability, poor aqueous solubility, and low bioavailability, encapsulation systems appear as an emerging and significant tool to overcome such issues. Microalgae bioactives can be greatly applicable in several areas, but without proper protection during processing and storage, as well as without suitable biopharmaceutical properties, the efficacy of their functionality may be utterly compromised.

In this review, we reported different published encapsulation systems incorporated with microalgae biomass, their extracts, or isolated bioactives to be applied in functional foods, disease treatment, and dermo-cosmetics. It was noteworthy that microalgae encapsulation for functional foods was the most investigated field, possibly due to the remarkable nutritional composition found in these microorganisms. However, many of the systems developed in this regard have proven their effectiveness in terms of stability and bioavailability improvement, suggesting they could also be applied to pharmaceutical or cosmetic purposes after in vitro and in vivo biological activity assessment.

For all the applications, the species *Haematococcus pluvialis*, followed by *Arthrospira platensis*, possess the highest number of studies using encapsulation systems, especially concerning their main bioactive compounds, astaxanthin and C-phycocyanin, respectively. Such bioactives are recognized by a strong antioxidant potential, which may provide positive effects in the treatment of several disorders. Accordingly, astaxanthin and phycocyanin are often considered and investigated as health-promoting substances in diverse areas, reflecting the number of scientific studies.

Overall, the beneficial effects of encapsulation over microalgae and their metabolites are evident. Through different encapsulation techniques, the studies have described that the presence of a coating material could, among others, promote stability enhancement under different storage conditions, allow suitable bioabsorption and skin permeation, improve physicochemical properties, and control the compound’s release in the target site. Furthermore, in vivo and in vitro evaluation of diverse biological activities have demonstrated that the encapsulated bioactives are able to provide a much higher therapeutic effect than their free form. Therefore, combining the biological properties of microalgae-derived compounds and the advantages of an encapsulation system seems to be a promising strategy to develop innovative and healthier products.

At the moment, there are only a few examples of commercial products that claim to contain encapsulated microalgae bioactives, mostly products based on astaxanthin from *H. pluvialis* [228,229] and carotenoids from the microalga *D. salina* [230]. However, it is expected that as studies on the scaling up of encapsulation processes and the evaluation of in vivo efficacy and safety progress, more and more companies will start launching new products.

## Figures and Tables

**Figure 1 marinedrugs-18-00644-f001:**
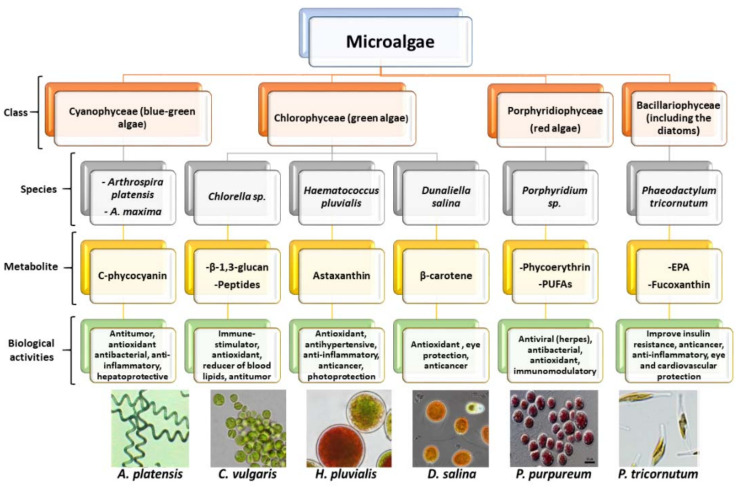
Main microalgae classes, their most important species, and associated biological activity (microalgae images were obtained from the “Microalgae strain catalogue”, second edition, published in the Enhance Microalgae Project, available at https://www.enhancemicroalgae.eu/wp-content/uploads/2020/05/EMA-Strain-catalogue-2nd-Edition.pdf).

**Figure 2 marinedrugs-18-00644-f002:**
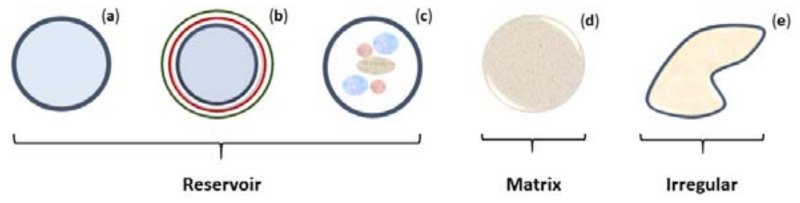
Different encapsulation systems structures: (**a**) mono-core, (**b**) multi-shell mono-core, (**c**) multi-core, (**d**) matrix and (**e**) representative of an irregular shape particle. Adapted from [111].

**Figure 3 marinedrugs-18-00644-f003:**
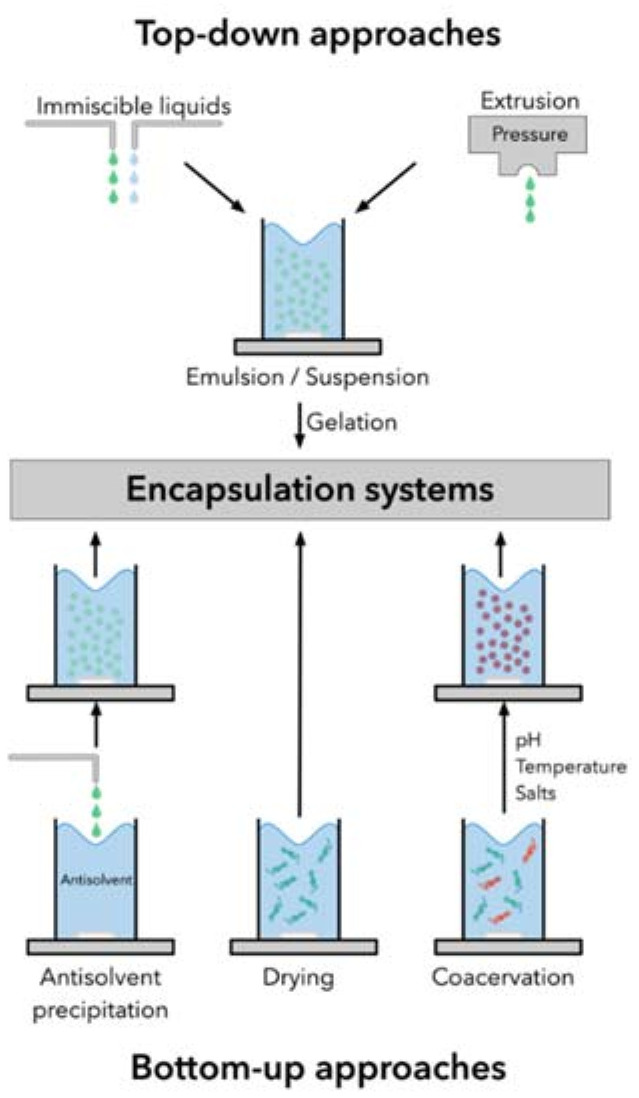
Encapsulation techniques of top-down and bottom-up approaches. Adapted from [99].

**Table 1 marinedrugs-18-00644-t001:** Literature review of encapsulation systems for food applications of the microalga *Haematococcus pluvialis*, its extracts, or bioactive compounds.

Core Substance	Coating Material	Encapsulation Technique/System	Application	Major Findings	Reference
Disrupted cells	Maillard reaction products	Spray dryer	Functional food	Stability improvement Better water dispersibility	[143]
Homogenized cells	Chitosan	Immersion	Functional food	Stability improvement under different storage conditions.	[144]
Astaxanthin or carotenoid extract	Polymerpoly(hydroxybutirate-co-hydroxyvalerat)(PHB)	Supercritical fluids (SEDS)	Functional food and pharmaceutical	Precipitation pressure had a higher influence on the formed particle size. Higher encapsulation efficiency is achieved when using higher biomass: dichloromethane ratio (10 mg mL^−1^) at the carotenoid extraction step.	[145,146]
Extract oleoresin	Capsul	Spray-dryer	Functional food	Thermal stability improvement.	[147]
Astaxanthin-enriched oil	Sodium alginate and low-methoxyl pectin	Vibrating-nozzle extrusion technology	Functional food	After one year of storage at different light, temperature, and oxygen conditions exposure, the microparticles were able to preserve the astaxanthin content ranging from 38% to 94%, with the highest result found when they were kept at lower temperatures.	[148]
Astaxanthin	Chitosan Carrageenan Calcium alginate	Nanoemulsion Extrusion	Functional food	Higher photoprotection was found for the nanoemulsion without polymeric coating. Chitosan beads provided higher protection to astaxanthin than alginate beads.	[149]
Lipid extract	Ulvan-pullulan	Electrospinning	Functional food	High encapsulation efficiency in the developed nanofibers, of around 90% for carotenoids and PUFAs. Promising protection of the lipid fraction of *H. pluvialis* encapsulated in a natural matrix composed of water-based polysaccharides.	[51]
Astaxanthin	Poly(ethylene oxide)-4-methoxycinnamoylphthaloylchitosan (PCPLC) Poly(vinylalcohol-co-vinyl-4-methoxycinnamate) Ethylcellulose (EC)	Polymeric nanospheres by solvent displacement	Functional food and pharmaceutical	Only PCPLC was suitable to form nanospheres. Greater improvement of astaxanthin thermal stability upon PCPLC nanoencapsulation.	[150]
Astaxanthin	Calcium-Alginate	Extrusion	Functional food and pharmaceutical	Temperature is the most influential environmental factor in astaxanthin degradation. Encapsulation improved astaxanthin thermal stability even after 21 days of storage at room temperature.	[151]
Astaxanthin oleoresin	Gum arabic and whey protein, alone or in combination with maltodextrin or inulin	Spray-dryer	Functional food	Microcapsules with 100% whey protein exhibited the highest colour and antioxidant stability. The turbidity retention of the microcapsules in aqueous dispersions depended on the pH and the carrier.	[152]
Astaxanthin oleoresin	Calcium-Alginate	External ionic gelation	Functional food	The diameter of oleoresin-loaded beads showed a strong dependence with alginate concentration and alginate/oleoresin ratio. Encapsulation yield was markedly affected by surfactant and alginate concentrations. The mathematical models developed can be used to predict the characteristics of natural astaxanthin-loaded microcapsules under different process conditions.	[153]
Astaxanthin	Soy phosphatidylcholine Cholesterol	Liposomes	Functional food	Astaxanthin exhibited a high retention rate in the liposomes after 15 days of storage at 4 °C. Cholesterol Enhancement of the antioxidant activity.	[154]
Astaxanthin oleoresin	Glyceryl behenate Oleic acid Lecithin	Nanostructured lipid carriers (NLCs) (melt-emulsification/ultrasonication technique)	Beverages (whey and non-alcoholic beer)	No astaxanthin loss and particle size growth were observed in the astaxanthin-NLCs-added whey after the storage time. Stability improvement of the NLCs in non-pasteurized CO_2_-free beer at low storage temperature. The organoleptic quality of NLCs-added beers was considered acceptable by the evaluators.	[155]
Astaxanthin oleoresin	Culled banana resistant starch Soy protein isolate	Emulsification	Functional food and pharmaceutical	Emulsions prepared with the starch-soy protein conjugate as wall material showed better physical and electrical stability compared to the one prepared only with soy protein. Stability improvement at different storage temperatures (6, 20, and 37 °C).	[156]
Astaxanthin	Poly (l-lactic acid)	Supercritical anti-solvent	Functional food and pharmaceutical	Stability improvement during 6 months of storage at different temperatures in comparison with free astaxanthin. Lower degradation rates were found at lower temperatures.	[157]
Astaxanthin oleoresin	Whey protein (WPI) Xanthan gum (XG)	Emulsification	Functional food	The addition of XG significantly increased emulsion stability in comparison to emulsions stabilized by WPI alone. Emulsified astaxanthin showed higher stability at lower temperatures during 15 days of storage. The combination of WPI-XG reduced the digestion and release of astaxanthin in comparison to the emulsion system stabilized by WPI alone.	[158]
Esterified astaxanthin	Whey protein Arabic gum	Complex coacervation	Functional food and pharmaceutical	Stability improvement and greater in vitro release rate compared to astaxanthin oleoresin. Enhancement of astaxanthin bioavailability.	[159]
Astaxanthin oleoresin	Precirol ATO 5 Stearic acid	Nanostructured lipid carriers (hot homogenization)	Functional food	NLCs were stable for at least 180 days at 4 °C and were capable of protecting astaxanthin antioxidant activity. NLCs exhibited in vitro capacity to protect human endothelial cells from ROS.	[160]
Astaxanthin extract	Sodium dode-cyl sulfate Decaglycerol monolaurate Decaglyc-erol monooleate	Microchannel emulsification	Functional food and pharmaceutical	O/W emulsion droplets remained stable at 25 °C with an encapsulation efficiency of over 98%, during 15 days storage period. The emulsification process was highly dependent on the emulsifier and extract types used.	[161]
Astaxanthin oleoresin	Maltodextrin Gelatine	Complex coacervation followed by spray dryer	Functional food	The microencapsulated astaxanthin maintained its antioxidant activity after spray drying, with higher values than vitamin C.	[162]
Astaxanthin	Modified lecithin (ML) Sodium caseinate (SC)	Nanoemulsion (high-pressure homogenization)	Functional beverages	SC-stabilized nanoemulsions showed good physicochemical stability (>70%) after 30 days of storage. Astaxanthin bioavailability was strongly influenced by the emulsifier used.	[163]
Astaxanthin	Blends of milk protein (whey protein isolate, or sodium caseinate) Soluble corn fibre	Spray dryer	Functional food	The developed microparticles demonstrated reasonably good water activity, surface morphology, encapsulation efficiency, and oxidative stability. Reconstituted emulsions showed good stability similar to the initial emulsions.	[164]
Astaxanthin	Tween 20 Whey protein isolate	Premix membrane emulsification	Functional food	The selected emulsification method was able to produce emulsions with remarkably narrow droplet size distributions. The astaxanthin emulsion was physically stable over 3 weeks of storage and it was able to preserve 70% of the astaxanthin content during this time.	[165]
Astaxanthin	Chitosan Salmon sperm DNA	Co-assembly	Functional food and pharmaceutical	Nanoparticles showed more powerful antioxidant activity than free astaxanthin, by improving the cytoprotective effect and ROS scavenging efficiency on H_2_O_2_-induced oxidative cell damage in Caco-2 cells. Enhancement of the cellular uptake efficiency.	[166]
Astaxanthin	Arabic gum Xanthan gum Pectin Methylcellulose	Emulsification- solvent evaporation	Functional food	Physicochemical characteristics of the nanodispersions were significantly (*p* < 0.05) influenced by the type and chemical structure of the polysaccharides used. Nanodispersions produced and stabilized with Arabic gum presented the smallest particle size and highest physical stability.	[167]
Astaxanthin	β-lactoglobulin Chitosan	Spontaneous self-assembly	Functional food	Stability and antioxidant activity improvement under acid treatment, high temperatures, and UV radiation. Chitosan coating was capable of providing a surface barrier to delay the release and degradation of astaxanthin in the gastrointestinal tract.	[168]
Astaxanthin oleoresin	Whey protein concentrate	Emulsification-Solvent evaporation	Functional food	Resuspended nanoparticles (NPs) in water exhibited superior stability than free oleoresin under extreme pH, high temperature, UV radiation, and metal-induced oxidation Simulated digestion of NPs showed high astaxanthin bioaccessibility.	[169]

**Table 2 marinedrugs-18-00644-t002:** Literature review of encapsulation systems for food applications of the microalga *Arthrospira*, its extracts, or bioactive compounds.

Core Substance	Coating Material	Encapsulation Technique/System	Application	Major Findings	Reference
Phycocyanin	Alginate Chitosan	Extrusion	Functional food	Thermal stability improvement of phycocyanin during storage when both polymers were used as the coating material. Microparticles were resistant to the acidic environment. Chitosan-coated alginate microparticles promoted a superior sustained release.	[184]
Phycocyanin	Maltodextrin Alginate	Spray dryer	Functional food	Phycocyanin microcapsules with alginate 0.6% and maltodextrin 9.4% showed the highest phycocyanin content, antioxidant activity, encapsulation efficiency, and blue intensity.	[185]
Phycocyanin	Calcium-Alginate	Ultrasonic and extrusion techniques	Functional food	Optimized conditions of C-phycocyanin ultrasonic encapsulation exhibited 98% entrapment efficiency when produced with 3% sodium alginate, 2.5% calcium chloride, and 1 mg/mL of C-phycocyanin concentration.	[186]
*A. platensis* biomass	Purified soybean phosphatidylcholine	Liposomes	Functional food	Liposomes subjected to homogenization presented a more uniform morphology and higher entrapment efficiency than the sonicated ones, demonstrating a possible strategy for *A. platensis*- loaded liposomes preparation.	[187]
Phenolic extracts	Rice and soybean lecithin	Liposomes	Functional food and pharmaceutical	Liposomes allowed the protection of phenolic compounds during the digestion gastric step, allowing the bioactive compounds to be released in the small intestine.	[188]
*A. platensis* powder	Maltodextrin pure or crosslinked with citric acid	Spray dryer	Yoghurt	Microencapsulated *A. platensis* presented higher thermal stability than its free form, showing better anti-inflammatory activity without exerting cytotoxicity. Yogurts added with encapsulated *A. platensis* presented a more homogeneous appearance. Microencapsulation was able to maintain the yogurt nutritional profile and improve the antioxidant activity throughout storage time.	[189]
Carotenoid extract	Yellow passion fruit albedo flour	Solvent displacement method	Functional food	The incorporation of the carotenoid extract showed high antioxidant activity, with carotenoids retention six times higher when compared to the nanodispersions containing synthetic β-carotene. The nanodispersions were able to keep the carotenoids’ stability for 60 days under refrigerated storage.	[190]
*A. platensis* powder	Alginate	Extrusion (internal and external ionic gelation)	Functional food	External gelation beads exhibited a more uniform and homogeneous morphology, as well as higher protein content when compared to internal gelation beads.	[191]
*Arthorspira* biomass	Calcium-Alginate	Spray dryer	Pasta	Microencapsulation was able to protect the biomass antioxidant potential in 37.8% from the pasta cooking conditions. The pasta properties were affected, but the overall acceptability index was not influenced by the addition of the microspheres.	[192]
*A. platensis* aqueous extract	Calcium-Alginate	Vibrational extrusion	3D printed cookies	Improvement of the antioxidant activity and colour stability of 3D printed cookies after processing and storage.	[136]
*A. platensis* extract	Pure trehalose or with maltodextrin	Freeze- and spray-drying Ball milling	Functional food	Freeze-dried samples, regardless of the matrix composition, exhibited the highest carrying capacity. The use of ball co-milling caused a complete degradation of the core compound during the applied processing. Delivery systems containing maltodextrin were more effective in preventing thermal degradation and preserving its colouring ability.	[193]
Phycocyanin	Polyvinyl alcohol	Electrospray	Functional food	Ultra-fine particles showed thermal resistance up to 216 °C, preserving phycocyanin antioxidant activity.	[194]
Phycocyanin	Glycerol monostearate and distearate Medium-chain triglyceride	Ultrasound-assisted high-shear homogenization	Functional food	Results showed that the type and amount of lipid and surfactant had a significant effect on the properties of the particle. Formulations prepared with glycerol distearate had the highest stability in terms of deposition and suspended particles.	[195]
Phycocyanin	Maltodextrin Κ- carrageenan	Spray dryer	Functional food	Results indicated that phycocyanin microparticles with 9% maltodextrin and 1% κ-carrageenan as coating materials produced the highest bulk density, particle size, and encapsulation efficiency.	[196]

**Table 3 marinedrugs-18-00644-t003:** Literature review of encapsulation systems for pharmaceutical applications of bioactives from the microalga *H. pluvialis*.

Core Substance	Coating Material	Encapsulation Technique/System	Application	Major Findings	Reference
Astaxanthin rich-extract	Cellulose acetate (CA)	Electrospinning	Antiaging	Astaxanthin extract-loaded CA nanofibers exhibited potential against oxidative stress of *C. elegans* with significant values. In vitro release assay showed a prolonged profiled. The stability of the nanofibers was significantly higher compared to that of the free extract under accelerated conditions.	[201]
Astaxanthin	Calcium alginate	Double emulsification	Hepato carcinoma	Microparticles formed had a good degree of roundness, dispersity, encapsulation efficiency, and pH responsiveness to avoid gastric degradation. Cellular studies demonstrated that encapsulated astaxanthin could inhibit hepatoma cells (HepG2 cell line) but it had relatively small or no impact on control hepatocytes (THLE-2 cell line).	[204]
Astaxanthin	Egg-yolk phosphatidylcholine	Liposomes	Lipoperoxidation inhibition	Astaxanthin strongly reduced lipid damage when different lipoperoxidation promoters were added simultaneously to the liposomes.	[205]
Astaxanthin	Methoxypolyethyleneglycol-polycaprolactone (mPEG-PCL) copolymer	Micelles (self-assembly)	Proliferation and differentiation of human mesenchymal stem cells	Mesenchymal stem cell (MSCs) differentiation results showed that 20 ng/mL astaxanthin-loaded polymeric micelles enhanced adipogenesis, chondrogenesis, and osteogenesis of MSCs by 52%, 106%, and 182%, respectively.	[206]
Astaxanthin	Cholesterol l-phosphatidylcholine	Liposomes	Hepatoprotection	Hepatoprotective and acute anti-inflammatory effects of the astaxanthin-loaded liposomes were confirmed by in vivo assay, being even superior to the positive control (N-acetylcysteine).	[202]
Astaxanthin	Cholesterol L-phosphatidylcholine	Liposomes	Antioxidant	Encapsulated astaxanthin activated more effectively antioxidant enzymes like superoxide dismutase, catalase, and glutathione S-transferase than its free form. Astaxanthin-loaded liposomes presented higher stability and in vitro bioavailability improvement when compared to the free molecule.	[207]
Astaxanthin	Ascorbyl palmitate	Nanoemulsion	Sublingual drug delivery	The developed nanoemulsion exhibited good uniformity dispersion and very low particle size. In vitro sublingual permeation studies showed that liposomes, together with this alternative route, are a promising alternative to enhance the bioavailability and therapeutic effect of astaxanthin.	[208]
Astaxanthin and α-tocopherol	Sodium caseinate	Nanoemulsion (spontaneous emulsification-ultrasonication)	Anticancer	Encapsulated astaxanthin was able to induce ROS generation and apoptosis through the apoptotic signalling pathway, in the nucleus and cytoplasm, as well as disrupt the mitochondrial membrane, in cancer cells.	[209]
Carotenoid-rich extract	Poly-lactide-co-glycolide	Polymeric nanocapsules by solvent displacement	Antioxidant	The carotenoids in their encapsulated form exhibited an antioxidant potential higher than the free extract and 9-fold higher when compared to ascorbic acid. The developed nanocapsules suspension when incorporated into a hydrogel showed a sustained release profile, with a higher release percentage when compared to the same formulation with the free extract.	[210]

**Table 4 marinedrugs-18-00644-t004:** Literature review of encapsulation systems for pharmaceutical applications of bioactives or extracts from *Arthrospira* species.

Core Substance	Coating Material	Encapsulation Technique/System	Application	Major Findings	Reference
C-Phycocyanin	Chitosan Xanthan gum	Coated liposomes (chitosomes)/spray and freeze-dryer	Colonic drug delivery (anti-inflammatory)	In vitro mucoadhesive study revealed that the spray-drying method is advantageous to prepare C-phycocyanin-loaded chitosomes with excellent mucoadhesive properties for colonic drug delivery.	[211]
Phenolic extract	Dimyristoylphosphatidylcholine Soybean asolectin	Liposomes	Antifusarium	The encapsulated extract exhibited higher antifungal activity (90% vs. 74%). and slower release profile when compared to its free form.	[212]
Phycocyanin	Sucrose laurate Sucrose erucate	Solid-in-oil nanodispersion	Transdermal drug delivery	In vitro skin permeation studies demonstrated that the nanodispersion with low surfactant content was able to facilitate the accumulation of phycocyanin in the *stratum corneum*, followed by its permeation into deeper layers for transdermal delivery.	[203]
C-Phycocyanin from *A. platensis*	Cholesterol l-phosphatidylcholine	Liposomes	Topical anti-inflammatory	In vitro skin permeation studies revealed that phycocyanin loaded-liposomes exhibited the best drug accumulation into the *stratum corneum* and in the whole skin, with higher amounts than the corresponding free phycocyanin gel. Liposomes were able to produce the same topical anti-inflammatory response as the free phycocyanin by using half the dose.	[213]
*A. platensis* fatty acids	Copper-alginate	Ultrasound emulsification	Anti-biofilm	Free fatty acids-loaded nanocarriers enhanced topical anti-biofilm growth activity with low concentrations (about 50% inhibition after 24 h at 0.1 mg/mL).	[214]
Aqueous extract	Polycaprolactone	Electrospinning	Tissue engineering (scaffold)	The addition of Arthrospira extract into PCL nanofibers showed potential to be a novel scaffold on fibroblast culture, improving the cells’ attachment rate and their infiltration depth.	[215]

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
