# Peer review of "Microalgae Encapsulation Systems for Food, Pharmaceutical and Cosmetics Applications"

_marinedrugs, 2020, doi:10.3390/md18120644_

Round 1
Reviewer 1 Report
Reviewer comments
In this review manuscript titled “Microalgae encapsulation systems for food, pharmaceutical and cosmetics applications” (Manuscript ID: marinedrugs-1020210), the authors Vieira et al. focuses on exploring encapsulation systems and types of encapsulation materials for microalgae derived products (e.g., biomass, extracts, bioactive compounds, etc.). However, I think if the authors add some improvements, the manuscript would deserve publication on Journal Marine Drugs once major and minor compulsory revisions are appropriately addressed.
Major Compulsory Revisions
- The authors could provide advantages and disadvantages of the encapsulation techniques/systems
- The authors could provide a critical analysis of the advantages and disadvantages of the encapsulation compounds.
- The authors could include microphotographies of some microalgae species
- The authors could provide chemical structures of the encapsulation compounds commonly used (e.g., alginate, synthetic polymers, etc.).
- The authors could provide fluxograms about the most widely used encapsulation techniques (e.g., spray drying, extrusion, etc).
Minor revisions
- Page 1 (lines 29-31) improve writing. Words biosynthesize and metabolize are redundant.
- Page 1 (lines 32-35) for the sentence “The use of microalgae by humans…of the world population”, please give the source for citation.
- Page 11 (lines 490-491) The sentence “Several papers have investigated…the carotenoid astaxanthin.” Maybe could change “papers” by “research groups”
Author Response
Reviewer 1:
Major Compulsory Revisions
1. The authors could provide advantages and disadvantages of the encapsulation techniques/systems.
R: The advantages of applying encapsulation systems are already described in Section 3. Firstly, overall advantages are reported in the paragraph at lines 305-314:
“The protective barrier provided by encapsulation offers several advantages. The primary reason to develop these particulate systems is to maintain the biological, functional, and physicochemical properties of the active agent [94]. The wall material serves as a protection from adverse environmental and processing conditions, such as the undesirable effect of light, temperature, moisture, and oxygen; therefore, contributing to an increase in stability and an extended shelf-life [95]. Another important advantage is the possibility to overcome challenges that normally restrict the incorporation of certain substances into commercial products. Encapsulation allows increasing the solubility of a compound into a dissimilar medium, masking unpleasant flavours, enhancing the bioavailability and bioactivity, as well as controlling and targeting the release of the core material as a response to external conditions (pH, temperature, etc.)”.
More specific advantages regarding the applications that are the focus of our manuscript can also be found in the last two paragraphs of this same section, in the lines 321-329 and 330-338. In addition, in Section 3.2, we have described the most widely used encapsulation techniques in the literature and, for each one of them, their main advantages and limitations.
2. The authors could provide a critical analysis of the advantages and disadvantages of the encapsulation compounds.
R: As we have mentioned in our manuscript, the choice of the encapsulating material for an encapsulation system depends on the desired properties of the final system and on the requirements of the encapsulated compounds (active agents). Different materials have been used in the literature for these purposes, but none of them can be assumed as universal for all types of active agents and applications. Thus, a polymer may be more advantageous for a certain application, but not for others.
Nonetheless, to improve the section text regarding the structure and compostion of the encapsulation systems, we have included the main advantages and limitations of choosing the different families of coating materials, namely biopolymers and synthetic polymers. The added information can be found in lines 367-369 and 378-381.
3. The authors could include microphotographies of some microalgae species.
R: As suggested, we have added microphotographs of the most studied microalgae species (included in Figure 1).
4. The authors could provide chemical structures of the encapsulation compounds commonly used (e.g., alginate, synthetic polymers, etc.).
R: We have included a table in the supplementary material (Table S1) with the most used coating materials for encapsulation purposes together with their CAS number, which together with a chemical database such as PubChem (https://pubchem.ncbi.nlm.nih.gov/) may provide any researcher with access to the structure and chemical properties. The CAS number can also be used for example in the ECHA database from the European Chemicals Agency (https://echa.europa.eu) to retrieve the safety data of registered susbtances.
5. The authors could provide fluxograms about the most widely used encapsulation techniques (e.g., spray drying, extrusion, etc).
R: We have accepted the reviewer’s suggestions by including a schematic representation of some of the most widely used encapsulation techniques as Figure 3.
Minor revisions
6. Page 1 (lines 29-31) improve writing. Words biosynthesize and metabolize are redundant.
R: The mentioned phrase was modified, as suggested.
7. Page 1 (lines 32-35) for the sentence “The use of microalgae by humans…of the world population”, please give the source for citation.
R: The entire paragraph between lines 32-38 provides information derived from references [3] and [4], already cited at the end of that paragraph. Nonetheless, to avoid any doubt, we have included both references at the end of the mentioned sentence.
8. Page 11 (lines 490-491) the sentence “Several papers have investigated…the carotenoid astaxanthin.” Maybe could change “papers” by “research groups”.
R: The sentence was modified, as suggested.
Reviewer 2 Report
This is a very good review. It is very rich and at the same time focused on a specific topic. Some parts are a little too long (for ex., details of encapsulation techniques), but this aspect is welcome to address an audience of non-expert in the field. The review is of high quality and in my opinion can be accepted for publication in the present form
Author Response
We thank the Reviewer 2 for his comments
Reviewer 3 Report
The manuscript "Microalgae encapsulation systems for food, pharmaceutical and cosmetics applications" by Vieira et al. presents a very interesting work and a comprehensive overview of the application of the microalgae encapsulation systems. In my opinion section Structure and composition should contain better characteristics of encapsulation structures (maybe some SEM photographs). In this section should also find the characteristics of liposomes and a description of encapsulation the microalgae metabolits.In my opinion section Encapsulation techniques describes the encapsulation methods. In my opinion, examples of encapsulated metabolites should be included.
And finally - are there any commercial preparations of the functional food, pharmaceutical or cosmetics with encapsulated microalgae?
Author Response
Reviewer 3:
The manuscript "Microalgae encapsulation systems for food, pharmaceutical and cosmetics applications" by Vieira et al. presents a very interesting work and a comprehensive overview of the application of the microalgae encapsulation systems. In my opinion section Structure and composition should contain better characteristics of encapsulation structures (maybe some SEM photographs). In this section should also find the characteristics of liposomes and a description of encapsulation the microalgae metabolites. In my opinion section Encapsulation techniques describes the encapsulation methods. In my opinion, examples of encapsulated metabolites should be included. And finally - are there any commercial preparations of the functional food, pharmaceutical or cosmetics with encapsulated microalgae?
1. Structure and composition should contain better characteristics of encapsulation structures (maybe some SEM photographs). In this section should also find the characteristics of liposomes and a description of encapsulation the microalgae metabolites.
R: The characteristics of liposomes and the advantages and disadvantages of this technique can be found in section 3.2 (encapsulation techniques), in lines 469-477. In this same section, there is further information about the encapsulated structures obtained using the different encapsulating techniques.
Regarding the description of the encapsulation of microalgae metabolites, an extensive literature review can be found in section 4, where we have reported the encapsulated metabolites, the coating material and encapsulation techniques used (including liposomes), as well as their application and main findings.
On the other hand, including the actual images of the capsules from the different cited papers would be very difficult. This is a sensitive issue, that usually involves requesting requires requesting permission / copyright clearance to each one of the different publishers. For this reason, we opted for including a diagram of the different types of structures produced (Figure 2), while the images can be consulted in the cited papers.
2. In my opinion section Encapsulation techniques describes the encapsulation methods. In my opinion, examples of encapsulated metabolites should be included.
R: Examples of different active agents that have already been encapsulated are mentioned in section 3.1 (Structure and composition), in lines 352-361. Regarding microalgae metabolites, which is the emphasis of this work, we have included an extensive number of examples in section 4 (Microalgae encapsulation), which also describes the associated encapsulation techniques used to obtain the final systems.
3. Are there any commercial preparations of the functional food, pharmaceutical or cosmetics with encapsulated microalgae?
R: Microalgae and their metabolites are widely marketed as nutritional supplements, but can also be found as functional ingredients to fortify various products. However, commercial preparations based on encapsulated microalgae are not reported in the literature, nor have we been able to find them in internet searches.
On the other hand, if we look for specific microalgae bioactives, it is possible to verify that some encapsulated preparations are already on the market. A few examples include two microencapsulated water-soluble H. pluvialis-derived astaxanthin powders (AstaPure® EyeQ, Algatech and BAIZ® Astaxanthin, Baiz) aimed to be readily incorporated into different commercial forms, such as tablets, cosmetics, beverages, and functional foods, or even a microencapsulated mix of carotenoids from the microalga D. salina, as a tablet grade powder (Betatene®, BASF).
This information was included into the manuscript (lines 890-894).